# Description of Light Environment in Broiler Breeder Houses with Different Light Sources—And How It Differs from Natural Forest Light

**DOI:** 10.3390/ani12233408

**Published:** 2022-12-03

**Authors:** Guro Vasdal, Kathe Elise Kittelsen, Fernanda Tahamtani, Dan-E. Nilsson

**Affiliations:** 1Norwegian Meat and Poultry Research Centre, Lorenveien 38, 0515 Oslo, Norway; 2Department of Biology, University of Lund, Sölvegatan 35, 22362 Lund, Sweden

**Keywords:** CFL, LED, poultry, production, welfare

## Abstract

**Simple Summary:**

As modern chicken production is based on keeping the birds indoors in buildings without windows, the artificial light environment is one of the key factors affecting their welfare and production. This paper aims to describe the light environment in nine broiler breeder houses with one of three different light sources (LED light, compact fluorescent lights or LED with UVA) using two different light assessments: spectrometer and the environmental light field (ELF) method. The paper also aimed to describe how the artificial light compares to the light in forest habitats. The results show that the light environments were relatively similar between the nine breeder houses and were typical for indoor environments. The artificial light environment differed significantly from the forest habitats, including a higher intensity even in the dense forests, a larger range of intensities in the natural environment (i.e., less bare) and a difference in the spectral balance between the forest and the breeder houses. The forests had roughly equal amounts of red and green, with a characteristic switch to more green than red in the upper part of the environment. The implications of these results, together with several potential improvements to the artificial light environment, are discussed.

**Abstract:**

Light is a key factor in poultry production; however, there is still a lack of knowledge as to describing the light quality, how to measure the light environment as perceived by birds, and how artificial light compares with the light in the natural forest habitats of their wild ancestors. The aim of this study was to describe the light environment in broiler breeder houses with three different light sources, using two different methods of light assessment. We also aimed to compare an artificial light environment with the light in a range of relevant natural forest habitats. A total of 9 commercial broiler breeder houses with one of three different light sources—Lumilux 830 CFL (*n* = 3), Biolux 965 CFL (*n* = 3) or LED Evolys with UVA (*n* = 3) were visited. Assessments of the light environment in the breeder houses were conducted using both a spectrometer and the environmental light field (ELF) method. ELF measurements from three forest types in south India (Kerala) were also included. We found that most aspects of the light environment were similar between the nine breeder houses and were not dependent on the type of light sources. The only clear difference related to the light source was the spectral balance, wherein 830 CFL had the most red-dominated light, 965 CFL had the most blue-dominated light and Evolys was intermediate but with more UV than the latter two. Plumage color had minimal effect on the light environment. Both the spectrometer and the ELF method provided valuable information. The spectrometer gave detailed values about certain aspects of the light environment, while the ELF described the light more in line with human and avian visual perception. We also found that the light environment in the investigated broiler breeder houses differs dramatically in all measured aspects from the natural light habitats of wild junglefowl, suggesting improvement possibilities in artificial lighting systems.

## 1. Introduction

Light is a key factor in poultry production, and one of the first studies on poultry lighting was published almost a hundred years ago [1]. Since then, extensive research has increased our understanding of lighting effects on poultry. Light plays an important role with regards to poultry behavior [2], production performance [3], sexual maturation [4], health [5] and welfare [6]. However, many of the studies on poultry lighting focus only on a single quality of the light environment, such as photoperiod [7], color temperature (CCT) [8], illuminance lux [7,9], light source [2], presence of UV [10] or presence of flicker [11]. 

Bird vision differs from human vision and is in many cases more advanced. The human eye has three types of retinal cone photoreceptors, peaking at 437, 533 and 564 nm, often referred to as blue-, green-, and red-sensitive cones, which gives humans a visible range between 400 nm to 700 nm. Birds have four distinct types of single cones and a double cone, and in *Gallus gallus domesticus*, the domestic chicken, the single cones peak at 415, 465, 540 and 600 nm [12], which gives them a visible range of at least 350 nm to 650 nm and theoretically makes them capable of distinguishing twice as many colors as humans [13]. One of the single cones is referred to as ultraviolet-sensitive [14], with sensitivity extending from 350 nm UV to 450 nm, allowing the bird to see a part of the spectrum that is invisible to humans (UV spectrum). Chicken eyes also contain rod photoreceptors, peaking in the green, and light-sensitive retinal ganglion cells, peaking in the blue [15]. Furthermore, the domestic chicken can also perceive light through extra-retinal photoreceptors in the brain [16], which can only be activated by long-wave radiation that penetrate the skull and impacts sexual development and maturity [7].

Over the last 100 years, different types of artificial light sources have been used in poultry facilities. Today, incandescent bulbs are being phased out in favor of more energy-efficient alternatives, such as compact fluorescent lamps (CFL) or light-emitting diodes (LED). LED lamps have been gaining popularity due to their low energy use, long operating life and availability in different wavelengths. However, due to the different wavelengths from different LED lamps, there are conflicting reports on the impact of LED lamps on poultry performance [17]. A study by Huth et al. [18] found that broilers housed under LED lights were less fearful and had better feed efficiency compared to broilers housed under CFL lights. However, two types of LEDs in the study did not have the same effect on the birds, which the authors attributed to the different spectra generated by the lights. Recently, light sources, including UVA and UVB, have been tested in poultry houses, and results point to positive effects, such as reduced fear and improved skeletal health [19]. However, more knowledge is needed about what specific quality of the light environment causes these effects.

Regardless of technological advances, the light environment in commercial poultry facilities differs considerably from light conditions in the natural habitats where the birds’ ancestors evolved and still reside. For instance, in forest habitats, the sunlight is filtered through green vegetation and thus differs from the light in open areas [20]. Wichman et al. [21] found that laying hens preferred ‘forest’ lighting (UV, RGB LED red:green:blue ratio simulating forest spectrums) over standard light (LED, 3000 K) and artificial daylight (UV, LED, 4500 K). Although light treatment resulted in only subtle changes in behavior, the birds were more active in daylight and forest light. There is a need to understand the differences between commercial light and natural forest light, as a more natural light can have positive effects on the birds’ behavior and welfare. 

The light environment is created by one or more sources, such as the sun or lamps. The light will hit objects and surfaces, and most of the light will reach the eye only after it has been reflected and transmitted by the environment. However, current methods used for measuring the light environment are often indirect for assessing the light reaching the eyes [22]. One such example is the lux meter, which measures the illuminance reaching the environment, usually from above, although most of the time light perception does not occur directly from the light source. Another commonly used light measurement is the spectrometer, which provides information on the spectral composition, illuminance, color temperature (CCT), color rendering index (CRI) and flicker index of the light environment. Thus, it provides high spectral resolution but low spatial resolution, whereas the eyes sample the visual world in quite the opposite way [23,24]. The spectrometer describes only a small part of the total light environment and in such detail that is far beyond the perception of the animal eye. 

In order to describe the light environment in a more biologically relevant way, Nilsson and Smolka [24] developed a new method to measure the light environment, called the environmental light field (ELF). The ELF quantifies biologically essential features, such as the radiance and photon flux, that characterize a light environment using a digital camera with a 180° fisheye lens. The ELF measures radiance as a function of elevation angle, which is similar to the way human and animal eyes function. This effect of elevation angle on light intensity is a neglected but fundamental feature of all light environments [25]. Furthermore, within each elevation angle of a scene, the light intensity varies to form the contrasts that are used for vision. These contrasts are another neglected variable that has a major and obvious impact on the experienced light environment. Furthermore, the spectral balance varies with elevation angle and constitutes an essential aspect characterizing the light environment. Thus, the ELF method measures environmental light more like the way human and chicken eyes perceive light. 

Although the ELF method might be biologically relevant, a modern spectrometer can be more available to farmers compared to an expensive digital camera with the need for statistical analyses afterwards, which is currently the case for the ELF method. However, how well the spectrometer and the ELF method correspond with each other in the same light environment is currently unknown. 

Broiler breeder production is a highly specialized and standardized process. Three of the common light sources in Norwegian broiler breeder flocks are Lumilux 830 CFL, Biolux 965 CFL and a LED light called Evolys consisting of LED bulbs and separate UVA bulbs, and the light environment created by these light sources is not described. Furthermore, two main colors of plumage exist in flocks of broiler breeders; white and brown, which may impact the light environment. 

The aim of this study was to describe the light environment in broiler breeder houses with three different light sources, including the effect of plumage color and to investigate the differences between different methods of light quality assessments. Finally, the aim was to compare the artificial light environment with different tropical forest light environments.

## 2. Material and Methods

### 2.1. Study Design

This study took place from October to November 2021. A total of 9 broiler breeder houses with one of three different light sources: Osram Lumilux 830 CFL(Munich, Germany), 58 W (*n* = 3), Osram Biolux 965 CFL (Munich, Germany), 30 W (*n* = 3) or LED Evolys with UVA (Type E21, Evolys, Oslo, Norway) (*n* = 3), were visited once during daytime when the lights were turned on. Comprehensive assessments of the light environment were conducted during each visit, first using a spectrometer and then the environmental light field (ELF) method [24]. 

This study did not involve any experimental procedures or handling of any animals. Therefore, approval by an ethics committee for animal experiments was not required according to Norwegian legislation [26].

### 2.2. Housing and Animals

The study included five flocks of brown plumage birds (Hubbard JA787) and four flocks of white plumage birds (Ross 308) (Table 1). All flocks consisted of around 7500 hens (range 7490–7575) and 550 roosters (range 490–672). All houses were similar with regards to physical factors, with concrete floor with wood shavings, automatic feeder and drinker lines, elevated slats and nest boxes, mechanical ventilation and artificial light. None of the houses had windows. Further details of the houses are presented in Table 1.

### 2.3. Data Collection

All 9 houses were visited when the lights were turned on, and no changes were made to the standard light program used by the farmer. In each house, two different types of measurements were conducted using a spectrometer (UPRTEK MK350S Premium, Elma Instruments, Oslo, Norway) in 10 random locations to cover all areas of the house (litter areas, near walls, center of the house, elevated slats and nest boxes). The first set of measurements recorded the light illuminance (lux), color temperature of the light source (CCT, expressed in *kelvin*, K), color rendering index (CRI, the effect of a light source on the color appearance of objects in comparison with a natural light source), light spectrum (the irradiance as a function of the wavelength emitted by the light source), flicker percent (the measure of the maximum vs. the minimum light in a cycle), flicker index (a measure of the quantity of light at high intensity against the quantity of light at low intensity over one cycle) and the number of oscillation of a light cycle in one second (hertz, htz). Mean values for each quality were then calculated per house. 

For the second set of determinations, the spectrometer was fitted with a Gershun tube, to reduce the angle of measurements from 180° to 30°, allowing the addition of UV data to the ELF measures. The spectral composition was measured 45° upwards and 45° downwards. A mean spectral curve for each of the two directions was calculated for each house. This was used to calculate the UV radiance reaching chickens from above and below the horizontal plane and in effect add a UV channel to the red–green–blue (RGB) channels of ELF measurements (see below). To relate the UV measurement to the calibrated ELF measurements, we computed the amount of light sampled by a chicken UV cone by multiplying the UV-cone spectral sensitivity [13] with the spectrometer curve and compared the area under the resulting curve with a similarly calculated curve produced by multiplying the spectrometer curve and the spectral sensitivity of the blue channel of the ELF camera [24]. In this way, we could relate the amount of UV light perceived by chickens to the blue channel of the ELF measurements. Even though this did not generate complete measurements for all elevation angles, it gave us two points (±45°) of UV that could be added to the ELF spectral graphs.

The ELF measurements were performed to quantify light as a function of elevation angle. We used a digital camera (Nikon 850D) (Nikon, Sendai, Japan) with 180° fisheye optics (Sigma 8 mm f/3.5 EX DG circular fisheye lens (Sigma Corporation, Bandai, Japan)), according to the methods described in detail in Nilsson and Smolka [24]. Two pictures aiming in opposite directions (to cover 360°) were taken in the same 10 random locations as for the spectrometer measures. Because the measurement method records radiances as a function of elevation angle, it was essential that the camera be kept horizontal during exposures. To facilitate camera orientation, we mounted a large bubble level on the camera. Exposures were taken after confirming that the bubble level indicated the horizontal orientation of the camera. An important aspect of this method is that it measures the total range of radiances at each elevation angle within a picture. To ensure a sufficiently large dynamic range, we used bracketing with three consecutive images, each exposed 3 full EV steps more than the previous, resulting in 60 images per house (10 locations × 2 directions × 3 different exposures).

### 2.4. ELF Analysis

The circular fish-eye images were remapped to an equirectangular projection with the elevation angle on the y-axis and the azimuth angle on the x-axis. From the remapped images, pixel values were extracted for each 3° band of elevation angle and converted to photon radiance values for red, green and blue pixels. The conversion was based on a calibration of the camera model involving the full range of exposure times, ISO settings and F-stops of the camera and its fish-eye lens. For each scene, the converted pixel values were used to calculate, as a function of elevation angle, the median radiance in red, green and blue, as well as white light (including all three spectral bands). Apart from median radiances at each 3° band of elevation angles, we also calculated the range of values including 50% and 95% of the pixel values at either side of each median value. This range, computed for white light, gives a measure of the image contrast and serves as a proxy for the amount of the spatial structure visible at each elevation angle.

Radiances are given as log_10_ values to provide manageable numbers over the full range of intensities that animal and human eyes are sensitive to. Using median rather than mean values returns “typical” radiances for each elevation angle, and any strong light sources are represented mainly as peaks on the upper bound of the 95% range. Details about the processing of the images to generate radiance data are described in Nilsson and Smolka [24].

To allow for the easy assessment and comparison of different light environments, we used the standard for the graphical presentation of ELF data [24,25]. The computed data was plotted in three different panels showing (1) the vertical gradients of median radiance (intensity) for white light, (2) the range of intensities (with 50–95% data range, normalized to the median radiance) and (3) the relative radiances (relative color) of red, green and blue light (normalized to the median white-light radiance) together with the two point measurements of UV (see above).

### 2.5. Comparative ELF Data from Natural Environments

For the comparison of light environments in the broiler breeder houses, we used ELF measurements from a large database of natural light environments [25]. The three selected measurements were from tropical forests in south India (Kerala, east of Thiruvananthapuram). These forests are inhabited by wild populations of gray junglefowl, *Gallus sonneratii*, which is the Indian counterpart of red junglefowl, *Gallus gallus*, from southeast Asia. Both species have contributed genetically to the modern domestic chicken [27]. Measurements from these forests were taken at daytime under sunny conditions. Spectrometer measurements from a similar tropical forest close to Cairns, Queensland, Australia, also taken on a sunny day, were used to add the UV band to ELF spectral charts in the same way as in the broiler breeder houses.

## 3. Results

### 3.1. Descriptive Qualities of the Light Environment—Spectrometer Results

There were relatively large variations both within and between the three different light sources, as measured by the spectrometer (Table 2). The three broiler breeder houses with 830 CFL had on average lower CCT (mean CCT: 2918.3 ± 15.98) meaning a warmer light with more red compared to the three houses with 965 CFL (mean CCT: 5349.63 ± 56.16) and the three houses with Evolys (mean CCT: 5313.73 ± 1305.15). Houses with Evolys had lower CRI (mean CRI: 74.80 ± 6.52) compared to 830 CFL (mean CRI: 83.25 ± 0.53) and 965 CFL (mean CRI: 93.07 ± 0.04). The houses with Evolys had the highest percent flicker (mean 8.67 ± 1.66), while houses with 965 CFL had the lowest percent flicker (mean 3.37 ± 0.28) (Table 2). 

The spectral composition (mean of 10 measures/house) in the three broiler breeder houses with 830 CFL can be seen in Figure 1. The spectrum of the 830 CFL had two sharp peaks at 540 nm (green) and 600 nm (orange to red). The spectral composition had similar curves in all three houses, but the amplitude was likely affected by the light intensity, resulting in higher peaks for house 2 compared to houses 1 and 3 (39.7 lux vs. 17.4 and 21.6 lux) (Figure 1).

The spectral composition (mean of 10 measures/house) in the three houses with 965 CFL can be seen in Figure 2. The spectrum of the 965 CFL had two lower peaks at 450 nm (blue) and 500 nm (blue) and two higher peaks at 550 nm (green) and 620 nm (orange). The spectral composition had similar curves in houses 5 and 6, but the low amplitude in house 4 was likely affected by the lower light intensity (7.4 lux in house 4 vs. 26.8 and 28.1 lux in houses 5 and 6) (Figure 2).

The spectral composition (mean of 10 measures/house) in the three houses with Evolys can be seen in Figure 3. The Evolys produces one sharp peak at 400 nm (violet) and a lower peak at 450 nm (blue). The spectral composition had similar curves in all three houses, but unlike houses with 965 or 830 light sources, the house with the highest light intensity (house 9, 30.5 lux) had the lowest amplitudes. 

A comparison of the spectral composition in all nine houses (mean of 10 measures/house) is presented in Figure 4. 

### 3.2. ELF Measurements of the Light Environments

In general, the light environments in all nine houses were relatively similar and quite typical for indoor environments with artificial lighting. The ELF results from houses with 830 CFL light source is presented in Figure 5, 965 CFL in Figure 6, and Evolys in Figure 7. Averages of the houses with each type of light source are compared in Figure 8. The charts in the panels B of Figure 5, Figure 6, Figure 7 and Figure 8 describe the amount of light (radiance) coming from different elevation angles, from straight up (top of diagram) to straight down (bottom of diagram), with the horizontal plane in the middle. The intensity axis is logarithmic, meaning that a radiance of 14 is 10 times brighter than a radiance of 13.

The absolute intensity, averaged over all elevation angles, differs between the houses but with no correlation to the type of light source. The light in houses 1, 4 and 7 were noticeably dimmer than in the other houses, whereas houses 2, 5, 6 and 9 were noticeably brighter, and 3 and 8 were intermediate. The difference between the brightest (2, 5, 6) and dimmest (4) is about 0.7 log units, or 5 times. In all houses, the intensity (radiance in panels B of Figure 5, Figure 6, Figure 7 and Figure 8) was higher above than below the horizontal plane. This may seem obvious, given that ceiling light fixtures are the only light sources, but since we use median values, the small angles looking directly into light sources have a negligible effect on the median radiances. The brighter upper field instead means that the ceiling material is brighter than the material on the ground.

**Figure 5 animals-12-03408-f005:**
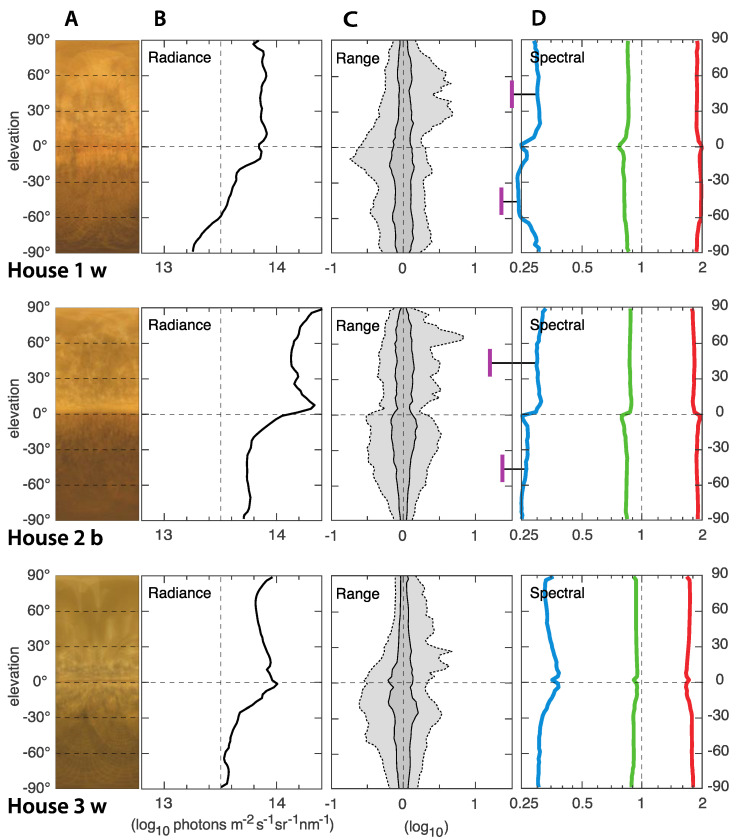
ELF measurements from the three houses with 830 CFL light sources with either brown (b) or white (w) plumage. Panels A are compressed average images, giving a direct impression of the light environment. The y-axis is the elevation angle in all panels. Panels B show median values of the absolute intensity (radiance) as a function of elevation angle. The intensity is in logarithmic units (the value 14 is 10 times brighter than the value 13). Panels C illustrate the range of radiances at each elevation angle. Dark gray contains 50% of the values around the median (at zero on the x-axis), and light gray contains 95%. Strong light sources generate peaks to the right (high intensity). The scale is logarithmic: −1 signifies 1/10 of the median intensity, and 1 signifies 10 times the median intensity. Panel D shows the relative contribution of red, green and blue (linear ratios on a logarithmic scale), as well as UV for two directions (45° above and below the horizon). The UV was measured separately with a spectrometer and computationally compared to the blue channel of the ELF camera. UV data is missing in house 3.

A general feature was a bright band around the horizontal plane, seen as a rightward peak on the radiance curves, and also directly seen in the average images (panels A in Figure 5, Figure 6, Figure 7 and Figure 8). Houses with brown-plumage birds (b) displayed a lower intensity just below the horizon, and a higher intensity just above the horizon compared to houses with white-plumage birds (w). However, this effect was relatively small.

**Figure 6 animals-12-03408-f006:**
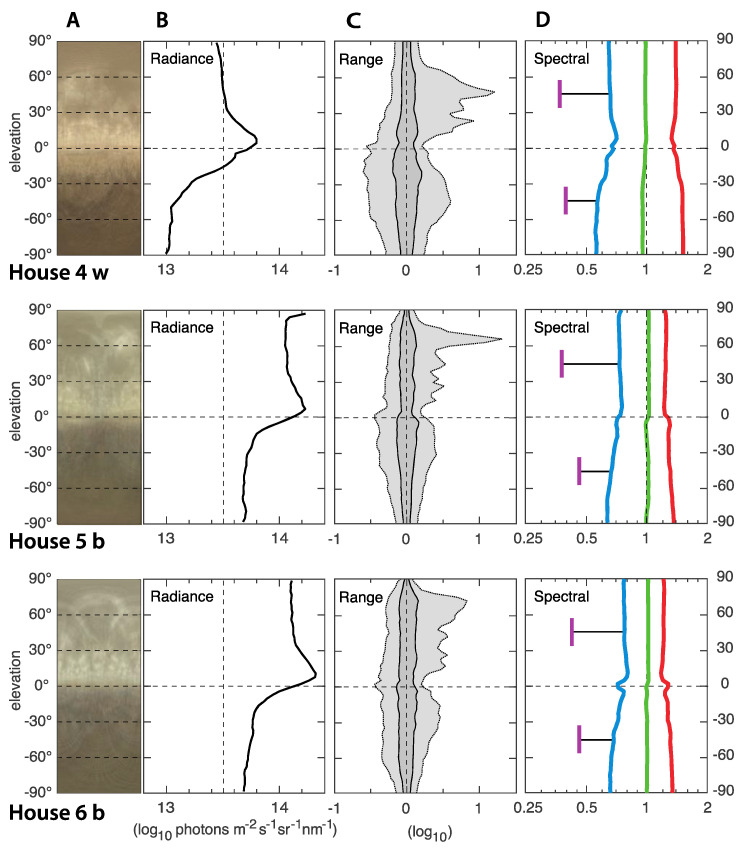
ELF measurements from the three houses with 965 CFL light sources with either brown (b) or white (w) plumage. Panels A are compressed average images, giving a direct impression of the light environment. The y-axis is the elevation angle in all panels. Panels B show median values of the absolute intensity (radiance) as a function of elevation angle. The intensity is in logarithmic units (the value 14 is 10 times brighter than the value 13). Panels C illustrate the range of radiances at each elevation angle. Dark gray contains 50% of the values around the median (at zero on the x-axis), and light gray contains 95%. Strong light sources generate peaks to the right (high intensity). The scale is logarithmic: −1 signifies 1/10 of the median intensity, and 1 signifies 10 times the median intensity. Panel D shows the relative contribution of red, green and blue (linear ratios on a logarithmic scale), as well as UV for two directions (45° above and below the horizon). The UV was measured separately with a spectrometer and computationally compared to the blue channel of the ELF camera.

The range of intensities, or contrast, describes the amount of visible structure seen at different elevation angles (panels C in Figure 5, Figure 6, Figure 7 and Figure 8). The dark gray band is a good general indicator of the amount of visible structure. Herein, this band was rather narrow at most elevation angles in all houses, meaning that the environment was rather featureless and bare. The light gray field of the C panels indicate the presence of very bright or very dark structures. In our measurements, this was generally larger below the horizon, where the field of view was dominated by other chickens. It was also larger for high intensities, above the horizon (jagged light gray area to the right), particularly in houses 4–6 (with 965 CFL light), indicating exposed light sources and a lot of direct light into the eyes of the chickens.

**Figure 7 animals-12-03408-f007:**
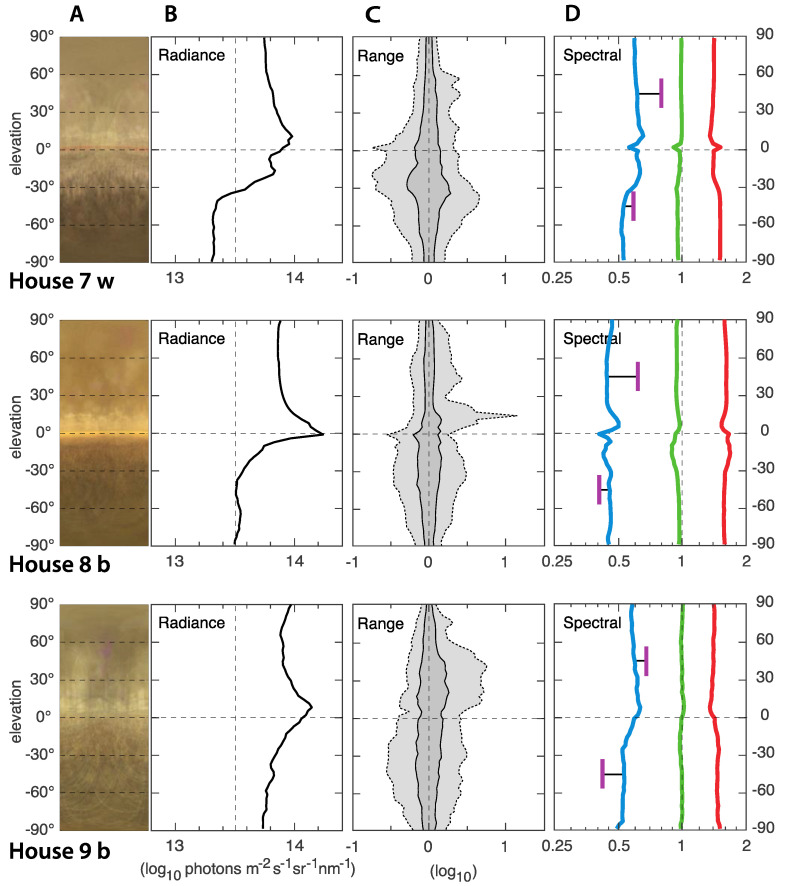
ELF measurements from the three houses with Evolys light sources with either brown (b) or white (w) plumage. Panels A are compressed average images, giving a direct impression of the light environment. The y-axis is the elevation angle in all panels. Panels B show median values of the absolute intensity (radiance) as a function of elevation angle. The intensity is in logarithmic units (the value 14 is 10 times brighter than the value 13). Panels C illustrate the range of radiances at each elevation angle. Dark gray contains 50% of the values around the median (at zero on the x-axis), and light gray contains 95%. Strong light sources generate peaks to the right (high intensity). The scale is logarithmic: −1 signifies 1/10 of the median intensity, and 1 signifies 10 times the median intensity. Panel D shows the relative contribution of red, green and blue (linear ratios on a logarithmic scale), as well as UV for two directions (45° above and below the horizon). The UV was measured separately with a spectrometer and computationally compared to the blue channel of the ELF camera.

**Figure 8 animals-12-03408-f008:**
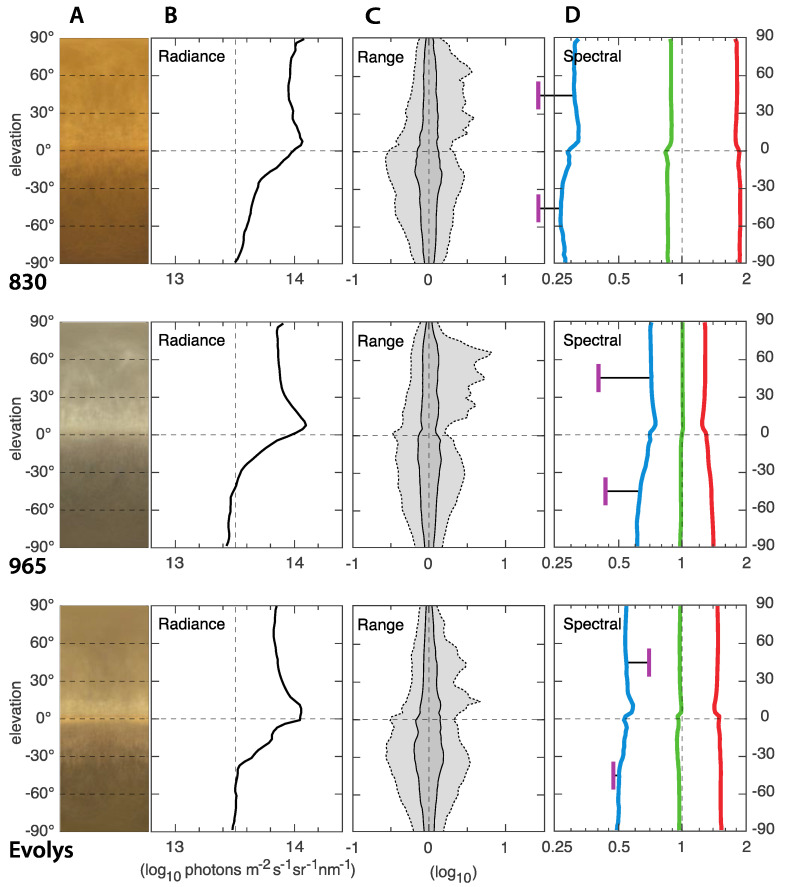
Average ELF plots for all houses with 830 CFL, 965 CFL, and Evolys light sources. Panels A are compressed average images, giving a direct impression of the light environment. The y-axis is the elevation angle in all panels. Panels B show median values of the absolute intensity [radiance] as a function of elevation angle. The intensity is in logarithmic units [the value 14 is 10 times brighter than the value 13]. Panels C illustrate the range of radiances at each elevation angle. Dark gray contains 50% of the values around the median [at zero on the x-axis], and light gray contains 95%. Strong light sources generate peaks to the right [high intensity]. The scale is logarithmic: −1 signifies 1/10 of the median intensity, and 1 signifies 10 times the median intensity. Panel D shows the relative contribution of red, green and blue [linear ratios on a logarithmic scale], as well as UV for two directions [45° above and below the horizon]. The UV was measured separately with a spectrometer and computationally compared to the blue channel of the ELF camera.

The spectral composition (light color balance) differs significantly depending on the type of lighting, but generally did not vary much with elevation angle. Houses with 830 CFL had the warmest light, with more than twice as much red light as green light and very little blue light (about a third of the amount of green light). The amount of UV was even lower. The coldest light was found in the houses with 965 CFL. Herein, the difference between red, green and blue was much smaller, but there was still more red and less blue. The relative amount of UV was lowest in houses with 965 CFL. Houses with Evolys had an intermediate balance between red, green and blue but much higher proportions of UV, which, above the horizon, was even more intense than blue. Comparing the houses with different light sources, the spectral balance stands out as the light environment feature that differed the most, depending on the type of light source.

### 3.3. The Artificial Light Environment Compared to Tropical Forest Light Environments

A comparison between light environments in the houses used for broiler breeders (Figure 8) and the natural forest environment of their wild ancestors (Figure 9) reveals a number of striking differences. First, the absolute intensities even in the dense forests (Dense and Regrowing, Figure 9) are about 1.5 log units or 30 times brighter than the chicken houses. In the semi-open forest, the light intensity was another log unit higher or 300 times brighter than in the chicken houses. Second, the horizontal plane was the darkest direction, quite the opposite of the chicken houses. Third, the range of intensities was much larger in the natural environment (it is far less bare). Finally, the spectral balance differed between the forest and the chicken houses. The forests had roughly equal amounts of red and green, with a characteristic switch to more green than red in the upper part of the environment. UV was less abundant in the forest compared to all chicken houses. The chicken houses that come closest to a tropical forest are those with 965 CFL, in particular houses 5 and 6.

## 4. Discussion

In this study, the light environment in broiler breeder houses with one of three different light sources was investigated. Our results show that the three houses with 830 CFL had warmer light (i.e., lower CCT) with more red compared to 965 CFL and Evolys, while 965 CFL had the coolest light (highest CCT). Measurements with the ELF method confirms the spectrometer findings that 830 CFL provides warm, red-dominated light, whereas the light from 930 CFL is colder and much less red-dominated. The spectral balance was very similar in houses with the same light sources but differed markedly between houses with different light sources. The spectral quality of the light can thus be attributed fully to the choice of light source. Longer wavelengths (red light) are known to stimulate the reproductive axis in broiler breeders [7], with positive effects on egg production [16] and on the reproductive traits of broiler breeder roosters [28]. On the other hand, green light may reduce egg production [16], while blue light generally promotes growth in young breeder birds [4]. Interestingly, Aviagen states in their breeder handbook that there is no need to provide broiler breeders with anything other than white light and that lamp type does not influence reproductive performance [29]. Further studies should focus on the potential effects of existing light sources on behavior and production in commercial breeder flocks. 

The spectral composition in the Evolys houses showed one sharp peak at 400 nm (violet), which was due to the UVA-producing LED sources. The presence of UVA is considered positive for poultry and has been shown to be preferred by laying hens [21] and to reduce fearfulness in both laying hens [30] and broilers [10,31]. In broiler breeders, UVA has been found to affect hens’ mate choice [32], likely due to UVA-reflective markings on the plumage of males. As more commercial light sources now contain UVA wavelengths, the effects on behavior, health and production in broiler breeders need to be studied further. The ELF measurements, aided with the computed UV readings from the spectrometer, reveal that the UV perceived by chickens was indeed present in both 830 CFL and 965 CFL houses, not only in Evolys houses. In 830 CFL houses, the UV was rather dim, but surprisingly, it was almost as bright in 965 CFL as it was with Evolys, where UV was expressly boosted.

Interestingly, the houses with Evolys had the highest percent flicker, with up to 16% flicker. Flickering lights are known to have detrimental effects on poultry [6]. However, the high flicker percentage in Evolys is compensated by much higher flicker frequencies of 200 and 400 Hz. Such high flicker frequencies will make flickering much less perceptible, because it is faster that the visual response of the photoreceptors in the retina [11]. Plumage color had minor effect on the light environment, and this effect could only be detected by the ELF method, wherein the spectral photon radiance had a higher intensity just above the horizon in the house with brown birds compared to white birds. 

Both the spectrometer and the ELF method provided valuable information about the light environment, with different advantages. The spectrometer is easily available to farmers and advisors compared to an expensive digital camera with the need for statistical analyses afterwards, which is currently the case for the ELF method. In addition to spectral information, the ELF method quantifies a wealth of other aspects of the visual environment that have previously not been considered. To assess these aspects and their potential importance, it is valuable to compare the artificial light environment with the light environments of wild junglefowl. The modern chicken has undergone lengthy domestication that has altered morphology and behavioral strategies, resulting in generally less active and less fearful birds that tend to allocate their resources towards production rather than behaviors that have a high energetic cost [33]. A comparative study on vision in junglefowl and modern chickens has not been done; however, it is likely that the birds retain some of the dependence on the light environments of their wild ancestors. The use of ELF data to compare light environments in breeding houses with that of their wild counterparts show that there are several differences between the light environments in breeder houses and natural forest habitats. The absolute light intensity is 30–300 times higher in natural habitats and angles around the horizonal plane are relatively darker in natural habitats but brighter in the breeding facilities. The intensity range is more than twice as large in natural habitats, i.e., there are far more visible structures, such as trees, than in the relatively bare breeding facilities, and the spectral balance differs markedly between the forest and the breeding facilities. It would be worthwhile to investigate if any of these differences can improve production in breeding facilities. The elevation-distribution of light, with a comparatively dark horizontal band, could be easy to accomplish with luminaire design, and the amount of spatial structure could be mimicked by patterned walls and ceilings. The spectral balance could also be easily manipulated by luminaire design. Our measurements show that 965 CFL comes closest to a natural spectral balance and that the 850 CFL is far too biased by long wavelengths [i.e., warm light], while Evolys boosted the amount of UV light above what is found in natural environments.

It is important to note that natural light environments are not static. During dusk and dawn, the intensity as well as the spectral composition changes gradually, and even during the day, light changes dynamically due to the movements of the sun and changing cloud cover. These continuous changes in the light environment are important in the behavioral control of all animals [25]. Feeding, resting and behaviors involved in reproduction are all necessary, and one cannot replace the other. In wild populations, it is essential that different behaviors are expressed at the time of day and under the environmental conditions when they are most successful [25]. This could be the reason why different amounts of red, green, blue and UV light have effects in broiler breeder facilities. Furthermore, in natural environments, the night has perceptible levels of light that could easily be produced in breeding facilities without much extra cost. 

## 5. Conclusions

In conclusion, the light environments were similar between the nine breeder houses and were not dependent on the type of light sources. Plumage color had minor effects on the light environment. The two assessment methods provide valuable information; the spectrometer gives detailed values about the light environment, while the ELF measures the light more like the way the animal eye perceives it. Finally, the light environment in the broiler breeder houses differs significantly from natural habitats and was quite typical of indoor environments with artificial lighting. In order to potentially improve production and welfare, several aspects of natural light should be tested out in commercial poultry housing.

## Figures and Tables

**Figure 1 animals-12-03408-f001:**
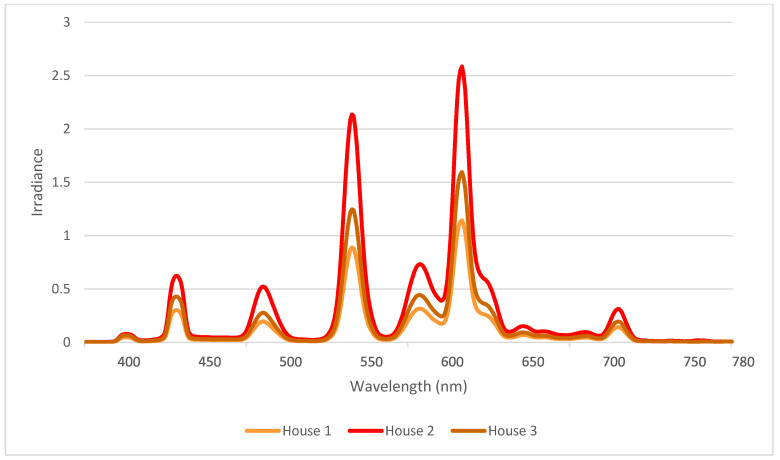
Spectral composition in the three houses with Lumilux 830 CFL light sources based on standard spectrometer measurements (mean of 10 measurements per house).

**Figure 2 animals-12-03408-f002:**
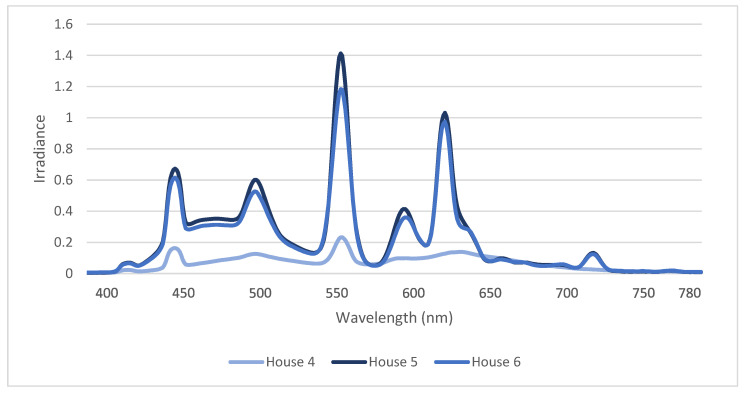
Spectral composition in the three houses with Biolux 965 CFL light sources based on standard spectrometer measurements (mean of 10 measurements per house).

**Figure 3 animals-12-03408-f003:**
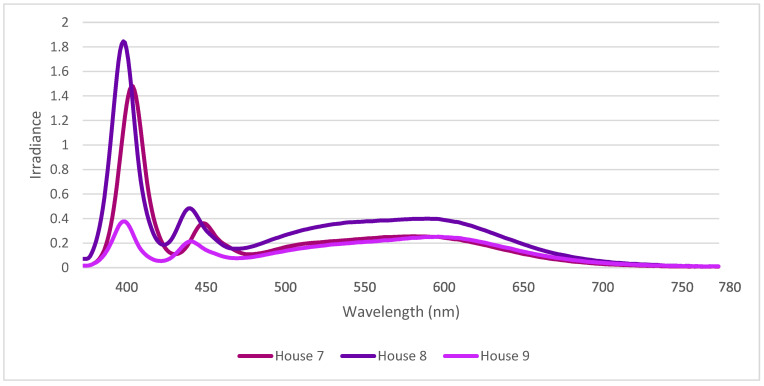
Spectral composition in the three houses with Evolys LED with UVA light sources based on standard spectrometer measurements (mean of 10 measurements per house).

**Figure 4 animals-12-03408-f004:**
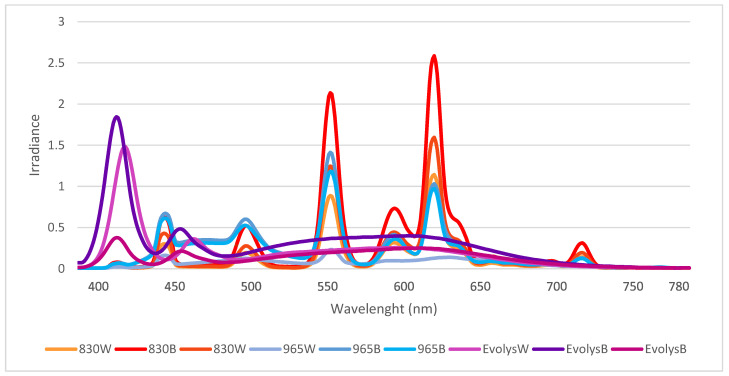
Spectral composition in all nine houses based on standard spectrometer measurements (mean of 10 measurements per house) in houses with 830 CFL, 965 CFL or Evolys light source and with either brown (B) or white (W) plumage.

**Figure 9 animals-12-03408-f009:**
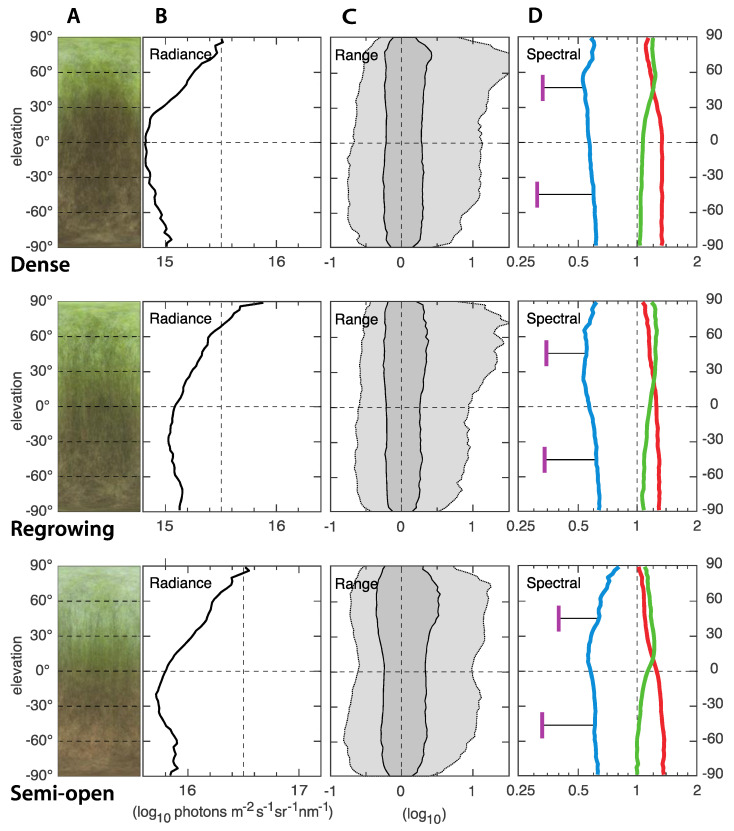
ELF measurements from the natural forest habitats of wild junglefowl in south India (Kerala). The different environments illustrate the range of habitats inhabited by wild junglefowl. The “Dense” is from a dense forest with multiple stories and a relatively open ground level. The “Regrowing” forest has fewer stories but dense ground-vegetation. The “Semi-open” forest has incomplete canopy coverage and a floor covered by dry leaf litter. The light environment in all three environments were measured under sunny conditions. Panels A are compressed average images, giving a direct impression of the light environment. The y-axis is the elevation angle in all panels. Panels B show median values of the absolute intensity (radiance) as a function of elevation angle. The intensity is in logarithmic units (the value 14 is 10 times brighter than the value 13). Panels C illustrate the range of radiances at each elevation angle. Dark gray contains 50% of the values around the median (at zero on the x-axis), and light gray contains 95%. Strong light sources generate peaks to the right (high intensity). The scale is logarithmic: −1 signifies 1/10 of the median intensity, and 1 signifies 10 times the median intensity. Panel D shows the relative contribution of red, green and blue [linear ratios on a logarithmic scale], as well as UV for two directions [45° above and below the horizon].

**Table 1 animals-12-03408-t001:** Details of light sources and house lay out in the 9 broiler breeder houses.

House	Light Source	Plumage Color	Bird Age at Visit [Weeks]	House Dimensions [m × m]	Ceiling Height [m]	Ceiling Shape and Color	No. Light Sources Ceiling	Wall Color
1	830 CFL	White	50	20 × 73	4.2	Concave, white	56 CFL	Light gray
2	830 CFL	Brown	42	24 × 74	8	Pitched, white	75 CFL	White
3	830 CFL	White	51	15 × 96	3.5	Concave, white	52 CFL	Light gray
4	965 CFL	White	47	15 × 95	3.1	Flat, white	72 CFL	White
5	965 CFL	Brown	49	15 × 85	4.7	Concave, white	66 CFL	White
6	965 CFL	Brown	34	14 × 81	3.2	Concave, white	48 LED	White
7	Evolys LED	White	60	21 × 71	4.2	Concave, white	48 LED 11 UVA	White
8	Evolys LED	Brown	46	15 × 82	3.5	Concave, white	42 LED 6 UVA	White
9	Evolys LED	Brown	34	14 × 81	3.2	Concave, white	36 LED 9 UVA	White

**Table 2 animals-12-03408-t002:** Spectrometer output (mean values ± SE of 10 measures per house) in the 9 broiler breeder houses.

House	Light Source	Plumage Color	Light Intensity (Lux)	CCT	CRI	Flicker Index	Flicker Percent	Htz
1	830 CFL	White D	17.467 ± 2.52	2707.8 ± 14.07	83.69 ± 0.16	0.01 ± 0.0	5.548 ± 1.68	100 ± 0.0
2	830 CFL	Brown V	39.786 ± 3.30	2798.0 ± 6.65	84.66 ± 0.23	0 ± 0.0	2.765 ± 0.23	100 ± 0.0
3	830 CFL	White S	21.62 ± 2.57	3249.2 ± 27.23	81.41 ± 1.22	0.002 ± 0.0	2.652 ± 0.39	100 ± 0.0
Average 830 CFL			26.29 ± 2.79	2918.33 ± 15.98	83.25 ± 0.53	0.004 ± 0.0	3.655 ± 0.76	100 ± 0.0
4	965 CFL	White G	7.446 ± 0.56	4698.3 ± 50.40	92.73 ± 0.07	0.01 ± 0.0	1.956 ± 0.08	100 ± 0.0
5	965 CFL	Brown G	26.81 ± 2.68	5713.3 ± 68.41	93.51 ± 0.03	0.01 ± 0.0	2.927 ± 0.12	100 ± 0.0
6	965 CFL	Brown K	28.18 ± 2.41	5637.3 ± 49.68	92.97 ± 0.04	0.01 ± 0.0	5.1 ± 0.65	100 ± 0.0
Average 965 CFL			20.812 ± 1.88	5349.63 ± 56.16	93.07 ± 0.04	0.0 ± 0.0	3.327 ± 0.28	100 ± 0.0
7	Evolys LED	White P	13.89 ± 3.65	7207.7 ± 3174.42	64.31 ± 11.05	0.049 ± 0.0	16.95 ± 2.31	400 ± 0.0
8	Evolys LED	Brown Ø	15.81 ± 2.66	3870.1 ± 90.05	85.73 ± 0.22	0.1 ± 0.0	4.31 ± 2.11	100 ± 0.0
9	Evolys LED	Brown R	30.55 ± 12.71	4863.4 ± 651.0	74.38 ± 8.29	0.01 ± 0.0	5.673 ± 0.56	100 ± 0.0
Average Evolys			20.08 ± 6.34	5313.73 ± 1305.15	74.80 ± 6.52	0.083 ± 0.0	8.67 ± 1.66	200 ± 0.0

## Data Availability

The original contributions presented in the study are included in the article, and further inquiries can be directed to the corresponding author.

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
