# Peer review of "Description of Light Environment in Broiler Breeder Houses with Different Light Sources—And How It Differs from Natural Forest Light"

_animals, 2022, doi:10.3390/ani12233408_

Round 1

Reviewer 1 Report

The work is very interesting and brings a lot of new elements to the current state of knowledge regarding lighting buildings for poultry with the use of various types of light.

The purpose of the work is clearly stated. The material used for the research is sufficient, the research methods have been selected appropriately.

The tables are clear.

The tables are easy to read. The results of the analysis of the light environment presented in the graphs are very interesting.

The conclusions from the conducted research are clear and result from the obtained research results.

Discussing the results against the background of other authors is very detailed.

The publications cited by the authors of the article are well selected. For the most part, the authors refer to the latest knowledge published in renowned scientific journals.

See more in the attachment.

Reviewer 2 Report

Dear Authors,

The theme and aim of your study seem interesting and promising. Unfortunately, I feel that your results are not sufficiently explored and given their right value. Please find below rather minor suggestions and requests for the correction of your manuscript, and use them as considered.

However, my major requirement is to improve your Discussion section which currently fails to sustain the importance of your research. I do believe that with more documentation your study can increase considerably its value for the reader in many aspects, and it can increase its practicality for the broiler industry. Also, please be aware of the differences between wild gallinaceous birds and domestic broilers while comparatively discussing/extrapolating their needs and the benefits of environmental influences (such as light) on their life quality and production (you can even state this as a limitation of your study).

Title: ‘the’ before ‘light environment’ is not compulsory

Please check if your paper does not need a Short Summary to comply with the Journal’s Instructions to Authors

Abstract

L15: the semicolon is not appropriate after ‘sources’, please consider the use of en/em dash or brackets for enumeration of the light sources

L16: please specify more what is meant by ‘light period’

L21-22: please keep the simple past tense all over the sentence

L24 and throughout the paper: please use simple past tense while describing own research (already performed)

L25: for style improvement consider replacing ‘more like the way the human and chicken eyes see it’ to ‘more similarly to human and avian visual perception’. However, I do have concerns regarding this wording, because of the differences between human and avian visual perception and even particularities of chicken vision within the general avian vision. Maybe writing ‘more similarly to chicken visual perception’ would be a better choice, but do we have enough scientifical proof to sustain the soundness of this statement?

L28: consider changing to ‘[…] differs dramatically in all measured aspects compared to the natural lighting habitats of wild jungle-flows, suggesting improvement possibilities of the artificial lighting systems.’

Introduction

Please check and follow closely the Instructions to Authors; correct your in-text citations throughout the manuscript

L45: citation needed at the end of the sentence

L51: please state the Latin name of the species instead of ‘chicken’ here

L53: please be more specific in stating what species are meant instead of ‘poultry’ here

L61: consider avoiding ‘But’ as a sentence opener (consider replacing with ‘However’, for example)

L67-70: please rephrase for more clarity. What had positive effects on (instead of ‘of’) welfare?

L67: consider changing ‘matching’ to ‘simulating’, ‘imitating’ or similar

L78: in order to avoid repetition of ‘behavior’ you may phrase that the chickens were more active

L80: consider changing ‘could’ to ‘can’

L81: consider using the plural (lamps or light bulbs)

L85: replace ‘measure’ with ‘measures’

L87: citation needed, or consider rephrasing (the ‘most will agree’). Suggested: ‘…most of the time light perception does not occur directly from the light source’

L88: replace with ‘…commonly used light measurement method is that by using the spectrophotometer…’

L91-92: consider rewording as ‘The spectrophotometer describes only a small part of the total light environment, and in such a detail that is far beyond the perception of the eye.’ Additionally, whose eye? Consider using ‘animal eye’ if both bird and human eye is emphasized, or be more specific if this was not the intended meaning.

L92-93: this sentence is redundant as it is (and not exactly academically written), please remove or complete it

L94-98: please avoid repetition (in order to measure […] a new method to measure […] called the ELF method. This method […] The method measures […])

L99: ‘structured’ may not be the best word here; it seems more about receptor function than structure

L105: please see the comment for L25

L107: consider to change ‘is more easily available’ to ‘can be more available’

L110-112: please move this sentence above (around L88) grouping it with the rest of the spectrophotometer description

L112: please try to avoid starting the sentence with ‘but’

L114-115: the second part of the sentence is redundant, as this information has already been given and it does not organically fit the following sentences

L118: please see the comment for L112

L121: please use simple past tense when referring to own research (already completed) including stating the study’s aim. Also, an impersonal voice is recommended in academic writing

L122: please insert ‘the’ after ‘including’

Materials and methods

L128: please consider avoiding brackets after brackets. Using a semicolon and including the n (should it be italicized?) in the same bracket as the light source type would be an option, to state that there were three poultry houses per lighting type, another

L131: please be more specific about the ‘light period’ for a better reader understanding

L132: ‘animal room’ is not an appropriate term; please consider removing it entirely

L135-136: numbers up to ten should be spelled out. For better fluency please consider ‘The study included five bird-flocks with brown plumage (…) and four with white feathers (…).’ In the next sentence please give the range (or standard deviation) for the number of birds per flock

L138: consider changing ‘feed’ to ‘feeder’

L145: if my understanding is correct, ‘first’ is appropriate here (but only in the following sentence). Please consider improving the text’s fluency. Suggestion: ‘In each birdhouse, two different types of measurements were conducted using a spectrophotometer (…), in 10 random locations, to cover all areas of the house (litter areas, near walls, the center of the house, elevated slats, and nest boxes). The first set of measurements recorded the light illuminance (lux), color temperature of the light source (CCT, expressed in kelvin, K), color rendering index (CRI, the effect of the light source on the color appearance of objects in comparison with a natural light source), the light spectrum (irradiance as a function of the wavelength emitted by a light source), the flicker percent (the measure of the maximum vs. the minimum light in a cycle), the flicker index (a measure of the quantity of light at high intensity against the quantity of light at low intensity over one cycle) and the number of oscillation of a light cycle in one second (hertz, htz). Mean values for each measurement were then calculated per house.’ Please include measurement units for each determination in the brackets to keep consistency.

L157: suggested: ‘For the second set of determinations the spectrophotometer…’

L158-159: the first part of the sentence becomes unnecessary if it is specified that the same measuring spots have been used for both sets of measurements

L163 and 169: ‘relate’ may not be the most appropriate term here. Please consider ‘to put in a relationship’, ‘to extrapolate’, or similar

L177: the use of the word ‘scenes’ and ‘recorded’ seems slightly confusing here, please be more specific (it suggests recording of video footage)

L178-179: instead of repeating the list of locations, consider stating that in the same 10 random places as for the spectrophotometric determinations or similar

L179-181: please move this sentence to the general description of the method (in the Introduction) or rephrase to concisely describe your method here. The word ‘scene’ needs a better definition for improved reader understanding (at its first usage)

L218-224: please include your motivation for selecting these specific data, such as scientific proof of the comparable visual ability of, or lighting effect on, the considered bird species and the domestic broiler breed, or any other reason as applies

Results

Figure captions: please pay attention to the use of ‘measure’, considering ‘measurement’ too

When presenting your results (of the study already finished), please use simple past tense consistently (e.g. when explaining the graphically presented results, 3.2 subsection, etc.)

Figure captions have to be complete, such as each figure and its caption to be comprehensible by the reader independently, as a standalone. Please correct incomplete captions (referencing another figure’s caption)

Figure 9 caption needs corrections: to keep consistency use “Dense” instead of ‘upper measurement’, “Semi-open” instead of ‘semi-open’; ‘All forests were measured’ is incomplete phrasing (i.e. the light was measured)

Discussion

Please rephrase the opener of this section instead of a word-by-word repetition of your study aim (already stated at the end of the Introduction section)

L362-369: please remove the plain repetition of the study results and rephrase them in light of the existing knowledge in the domain

L378: include citation (reference list number in square brackets, as requested by the Journal) after the name of the author instead of stating it at the end of the sentence

L401: include the species

L404: please be consistent regarding the use of American and British English (e.g. color/colour, behavior/behaviour, etc.)

L404-405: citation needed. Also, please extend the discussion on ‘other effects on behavior

L409-415: please do not repeat information previously stated in the paper

L412: citation needed. Expand the explanation of your reasons for this statement

L423: explain more the reasons for which the aspects could be more suitable for domesticated chickens

L425: please take into consideration the effect of adaptation of birds to their environment and include this aspect in your discussions

L427: citations are needed

L428: what do you mean by ‘structure’?

L430-431: repeatedly stating that more studies are needed can become ‘annoying’ to the reader and lowers the value of your study, in my opinion

L432-434: if these statements are made as suggestions, please consider replacing ‘is’ to ‘can/could be’ or similar

L434: biased ‘to’ or ‘by’?

L437: please consider avoiding personification in academic writing (Evolys have overdone it)

L443: but or and?

L445: citation needed

L450: please see the comment for L430

L450: the Conclusions should be a separate section; please follow closely the Instructions for Authors

L450: please be more explicit about what you mean by differences within the three types of light sources

L451: ‘marginal’ effect may not be the proper term

L452: please use the simple past tense, avoid repetition of previous statements, and see comment for L430

References list

Please correct this list according to the Journal’s Instructions to the Authors.

Reviewer 3 Report

Good research for the poultry breeder industry and well written manuscript.

Line 11: can delete "the" in front of artificial

Page: 2

Line 73 can delete "the"

Page: 3

Line 122: delete we also want to,

Line 124: Delete want to

Line 137: remove space in 7500

Page: 9

Line 271: change are to were

Round 2

Reviewer 2 Report

Dear Authors,

 Thank you for this enhanced version of your manuscript. I am glad to see the improvements you have made regarding both scientific content and form, to offer the best possible experience for the reader.

There are a few minor copy-editing aspects I would like to see resolved; I list them briefly below:

-          a few empty lines to be removed;

-          numbering the parts of the paper (as per the Journal’s instructions);

-          consistent formatting of the subtitles (most are written in italic, but not all), and numbering them;

-          beginning the title with a capital letter for Table 1; removing the hyphens between the table numbers and their titles;

-          correcting ‘10 measures’ to ’10 measurements’ in figure titles;

-          using simple brackets instead of square ones in figure captions;

-          following closely the Journal’s instructions to format the Reference list (remove the spaces between the initials of the author’s first and middle names, abbreviate journal titles, italicize journal issue numbers; format the references at no. 26 and 29 properly, with chapter title included, and I would give the English title for ref 26 too; keep consistency by adding dots at the end of each reference-list entry).

Indeed, I did not mention these aspects at the first review round, but they are very minor; may be components of the proofreading process rather than that of a review. Thus, please consider my review a positive one, with no real correction request.
